# Identification of the *SUT* Gene Family in Pomegranate (*Punica*
*granatum* L.) and Functional Analysis of *PgL0145810.1*

**DOI:** 10.3390/ijms21186608

**Published:** 2020-09-10

**Authors:** Krishna Poudel, Xiang Luo, Lina Chen, Dan Jing, Xiaocong Xia, Liying Tang, Haoxian Li, Shangyin Cao

**Affiliations:** Zhengzhou Fruit Research Institute, Chinese Academy of Agricultural Sciences, Zhengzhou 450009, China; krishnapoudel08@gmail.com (K.P.); luoxiang@caas.cn (X.L.); 82101171109@caas.cn (L.C.); 82101176054@caas.cn (D.J.); xiaxiaocong12@163.com (X.X.); 82101185030@caas.cn (L.T.); lihaoxian@caas.cn (H.L.)

**Keywords:** genetic transformation, transgene, cloning, expression analysis, cytological observation

## Abstract

Sucrose, an important sugar, is transported from source to sink tissues through the phloem, and plays important role in the development of important traits in plants. However, the *SUT* gene family is still not well characterized in pomegranate. In this study, we first identified the pomegranate sucrose transporter (*SUT)* gene family from the whole genome. Then, the phylogenetic relationship of *SUT* genes, gene structure and their promoters were analyzed. Additionally, their expression patterns were detected during the development of the seed. Lastly, genetic transformation and cytological observation were used to study the function of *PgL0145810.1*. A total of ten pomegranate *SUT* genes were identified from the whole genome of pomegranate ‘Tunisia’. The promoter region of all the pomegranate *SUT* genes contained myeloblastosis (MYB) elements. Four of the *SUT* genes, *PgL0328370.1*, *PgL0099690.1*, *PgL0145810.1* and *PgL0145770.1*, were differentially expressed during seed development. We further noticed that *PgL0145810.1* was expressed most prominently in the stem parts in transgenic plants compared to other tissue parts (leaves, flowers and silique). The cells in the xylem vessels were small and lignin content was lower in the transgenic plants as compared to wild *Arabidopsis* plants. In general, our result suggests that the MYB cis-elements in the promoter region might regulate *PgL0145810.1* expression to control the structure of xylem, thereby affecting seed hardness in pomegranate.

## 1. Introduction

Pomegranate (*Punica granatum* L.) is one of the oldest fruits known to mankind, and it thrives better in tropical and sub-tropical regions [1]. It used to be placed in the monogeneric family Punicaceae [2] but now has been shifted to family Lythraceae by the Angiosperm Phylogeny Group [3] on the basis of phylogenetic studies. The fruits have a medicinal value and are a good source of compounds like antioxidant phenolics and anthocyanins, which depends upon the diversity of the pomegranate genotypes. The fruit is regarded as a “superfruit” for its health benefits [4]. This has created awareness globally for an increase in the production area as well as the demand for consumption [5,6]. The prominent increase in the consumption of the fruits has increased the demand for the fruits in the market, which encourages researchers and breeders to develop and breed ideal types of varieties [7], such as a soft-seeded cultivar. In the past, breeding fruit crops was traditional and time-consuming as the fruit crops have a long generation time, large size and a long juvenile phase for seedlings, and it takes a long time to obtain the fruits ready for marketing [8,9]. Mapping the soft-seeded genes and adopting marker assisted selection (MAS) will accelerate the progress of soft-seeded cultivar breeding.

Although there are some research works on seed hardness traits in pomegranate, there are no clear molecular mechanisms or identification of genes responsible for seed hardness. The seed coat of pomegranate contains high levels of lignin [10]. The lignin content in the seed coat has a positive correlation with the characteristics of seed hardness, and amount [11,12]. Based on the transcriptomic analysis, WRKY, AP2 like, MYC, MYB and NAC transcription factors (TFs) are expressed differently in soft- and hard-seed pomegranate varieties [13,14]. The miRNA–mRNA influence seed hardness in pomegranate by acting on the cell wall. NAC1, WRKY and MYC are involved in seed hardness due to differentially expressed mdm-miR164e and mdm-miR172b. The miRNA–mRNA network results in a complex biological process [15]. This cell wall biosynthesis and degradation might play a role in seed hardness in pomegranate [16,17]. In another study, between “Tunisia” soft-seeded and “Sanbai” hard-seeded varieties, a single nucleotide polymorphism (SNP) (T-C) at the 166bp position was non-synonymously substituted with lysine and replaced by glutamic acid. The lignin biosynthesis and seed hardness in pomegranate is due to the NAC transcription factor, *PgSND1-like* gene [18].

Sucrose plays a key role in plant growth, development and crop yield [19,20,21,22]. *SUT*/*SUC* genes have a key role for transporting sucrose from source to sink organs. The sucrose produced in the source leaves is the major photosynthetic product that is transported to the storage organs (sink tissues), roots, stems, flowers, fruits and seeds, in the plants through the vascular tissue [23,24,25,26,27]. It is regarded as the major product that is necessary for the growth and development of the plants [28,29,30]. Depending on the plant species, sucrose is transported in different ways in the plant system to the phloem, apoplastically over short distances through the plasma membrane and symplastically through plasmodesmata [31,32]. Apart from short distance loading, sucrose is transported to sink organs, which is the final consumption location, and storage organs in plants [29,30,33,34]. The sucrose transporter genes regulate the transport activities to adapt to changes in the external environment, such as temperature, photoperiod, pathogens, etc. [35,36].

The first *SUT* gene in plants, *SoSUT*, was reported from spinach [37] then, a large number of *SUT* genes were found in rice [38] and *Arabidopsis* [39,40,41]. The translocation of sucrose from the leaves to the seeds plays an important role in the maximization of the yield in maize [42]. Similarly, in cotton during the seed development stage, sucrose is transported to enhance fiber formation, development and elongation [43]. Additionally, in rapeseed, *BnA7SUT1* was significantly associated with the number of effective branches, siliques per plant and seed weight [44]. In an antisense transformation study in a tomato crop, the *SUT* genes played an important role in the retardation of sucrose translocation, which reduced the fruit size and even the fertility was lowered [45,46]. Basic research and crop improvement activities in pomegranate crops are limited [47]. However, there is no information about the *SUT* gene family in pomegranate. We have no idea about the function of *SUT* genes, and the relationship between the *SUT* genes and soft-seeded phenotypes. Therefore, the present study was designed to identify the *SUT* genes and analyze their functions in the development of seed hardness in pomegranate.

## 2. Results

### 2.1. Identification of SUT Gene Family

Through sequence analysis from the “Tunisia” pomegranate genome database, a total of 10 members of the *SUT* gene family were obtained on Chr1, Chr3, Chr4, Chr5, Chr6 and Chr7 (Table 1). Among them, the *SUT* genes were distributed more (three each) in Chr3 and Chr6. The analysis of the results shows that the gene length varies greatly, with an average length of 4406.7bp. The maximum gene length was observed in *PgL0237030.1* with 8014bp and the minimum in *PgL0281810.1* with 279bp. The highest and the lowest differences among the longest and the shortest sequences of *SUT* genes of pomegranate are 7735bp and 158bp. The average adenine, cytosine, guanine and thymine contents in the pomegranate *SUT* genes were 28.50%, 21.58%, 22.77% and 27.15%, respectively. However, in the promoter sequences, the average adenine and thymine content was higher than that of the gene sequences. The average adenine and thymine contents in promoter sequences were 30.51% and 30.71%, which are higher by 7% and 11.31%, respectively. Meanwhile, the average cytosine, guanine and GC contents were 19.17%, 19.62% and 38.79% in the promoter sequences, which were lower than those of gene sequences (Table 2).

We analyzed the *SUT* proteins and their amino acid composition and found that the protein sequence length varied by 92–1251 amino acids (Appendix A). Meanwhile, the variation in molecular weight was 10260.37–142472.24 Dalton. The proteins of *PgL0281820.1* and *PgL0281800.1* had no tryptophan and the protein of *PgL0281810.1* was without cysteine acid. In general, the average leucine content of the *SUT* protein sequences was the highest at 10.69, and the average cysteine content, 0.82, was the lowest. These structural features of the genes and proteins may relate to the functional diversity among the *SUT* genes.

### 2.2. Phylogenetic and Gene Structure Analysis of Pomegranate SUT Genes

From the phylogenetic tree analysis, we conclude that the pomegranate *SUT* genes are mainly divided into three major categories: group1, group2 and group3 (Figure 1). Group1 and group3 contained four and five *SUT* genes, respectively. *PgL0237030.1* forms a separate branch. Among all the *SUT* genes, *PgL0237030.1* from group2 contained the largest number of exons and introns, followed by *PgL0099690.1* and *PgL0233780.1* from group3. Meanwhile, in the other *SUT* genes from group1, there are fewer than ten exons and introns. Thus, the variation of the number of exons and introns was largely in accordance with the structure categories.

### 2.3. Promoter Analysis of Pomegranate SUT Genes

PLACE analysis revealed that promoter regions of all the pomegranate *SUT* genes contained multiple MYB elements (Figure 2). These MYB elements are mainly MYB2AT, MYB2CONSENSUSAT, MYBCORE, MYBCOREATCYCB1, MYBGAHV, MYBPLANT, MYBPZM, MYB1AT and MYB1LEPR. Among the ten *SUT* genes, *PgL0145810.1*, *PgL0237030.1* and *PgL0181920.1* contained more than twenty MYB elements and *PgL0099690.1* contained the least, with seven MYB elements.

### 2.4. Expression Analysis of the SUT Genes in Seed Development

Previous transcriptomic data [13,14,15] showed that the expression of four *SUT* genes, *PgL0328370.1*, *PgL0099690.1*, *PgL0145810.1* and *PgL0145770.1*, may be involved in regulating the development process of pomegranate seeds (Appendix A). A qRT-PCR analysis further proved that the relative expression levels of *PgL0145810.1* were normally higher in the mature seed of soft-seeded varieties (ZSL8, Yi3 and Tunisia) compared to the hard-seeded varieties (Jiu Zi Hong, Yi2000-1 and Sanbai) (Figure 3A; Appendix A). The result indicated that *PgL0145810.1* may negatively regulate the development of seed hardness in pomegranate.

### 2.5. Growth Phenotype and Expression of PgL0145810.1 in Transgenic Lines

To further investigate the function of *PgL0145810.1,* we cloned *PgL0145810.1* and dipped the flower buds of *Arabidopsis* in an *Agrobacterium* cell suspension. The transgenic *Arabidopsis* lines were confirmed through qRT-PCR analysis (Figure 3B; Appendix A). The transgenic lines were grown for three generations (T3) to obtain homozygous plants for the transferred gene.

Growth phenotypes of the *PgL0145810.1* transgenic lines were highly different from the wild type in terms of plant height, number of leaves, number of stems and number of seeds per siliques (Figure 4C; Appendix A). Additionally, the growth phenotypes of the *PgL0145810.1* transgenic line L4 was comparable to that of L12. Further expression analysis indicated that the expression level of the *PgL0145810.1* gene was more prominent in the stem parts compared to other tissue parts (leaves, flowers and silique) either in L12 or L4 (Figure 4B; Appendix A). However, the expression levels of *PgL0145810.1* in the stem of L12 and L4 were higher than that of the wild type. This result is similar to the observation that the lignin content in the transgenic *Arabidopsis* plant L12 was reduced by 13.63% and in L4 by 9.37% as compared to the wild type (Figure 4D). Thus, the expression level of *PgL0145810.1* is related to the stem differences between wild type and transgenic lines.

### 2.6. T-DNA Insertion Site in Transgenic Line

Genome walking was performed to locate the insertion site of T-DNA in the transgenic lines. The sequencing of the flanking genomic sequence showed that the T-DNA insertion site in transgenic lines was at Chr5: 4809575–4810941. The T-DNA insertion site was located in the coding sequence of the gene located in Chr5: 4808091–4811093, which was annotated as cyclic nucleotide-gated channel 18 (CNGC18), regulating the essential role in pollen development and pollen tube growth (Appendix A).

### 2.7. Cytological Observation of the Stems

To further assess the function of *PgL0145810.1,* cytological observation was conducted in the tissue of stems from the thirty-five-day-old seedling stage collected from WT, L4 and L12 plants (Figure 4A). We observed that the numbers of the cells in the vascular bundle and pith were significantly higher, while the size of the cells was comparatively small in the transgenic lines L4 and L12 as compared to the WT plants (Figure 4E–G). Similarly, the length and the thickness of the xylem vessel in both of the transgenic lines, L4 and L12,were shorter and reduced as compared to the wild type plants, while the expression patterns of *PgL0145810.1* in the stems of L4 and L12 were higher. Collectively, this implies that high expression of *PgL0145810.1* regulated the structure of the xylem.

## 3. Discussion

The present study first explored all the *SUT* gene family members that are present in the pomegranate genome and identified a total of ten *SUT* genes. Phylogenetic analysis divided the pomegranate *SUT* genes into three major categories, and variations of the number of exons and introns were largely in accordance with the structure categories. Additionally, *PgL0328370.1*, *PgL0145810.1* and *PgL0145770.1* were proved to relate the seed development by the transcriptome, and all of them came from group1. Thus, the structure group of the *SUT* genes may determine gene structure and biological diversity in pomegranate.

Sucrose is the primary photosynthetic product that is translocated through the phloem in most economically important plants [48,49] from source to sink organs and constitutes a key component for carbon portioning for the whole plant [50,51,52]. *SUT*/*SUC* is a large gene family [53] and plays an important role in the biological evolution and the formation of the important traits [28,54] such as flowering, plant height and crop yield [19,20,21,22]. The rapid development of high-throughput sequencing and bioinformatics analysis technology has made transcriptome analysis easy for researchers to detect the genes associated with complex quantitative traits [55,56]. Through transcriptomic analysis, NAC1, WRKY, MYB and MYC transcription factors were proved to be responsible for the formation of seed hardness in pomegranate [13,14]. Our study first reveals that the expression of *SUC* genes (*PgL0099690.1*, *PgL0145810.1*, *PgL0145770.1* and *PgL0328370.1*) is involved in the development of seeds in pomegranate. In particular, the down regulation of *PgL0145810.1* is related to an increase in the seed hardness trait in pomegranate. Thus, we inferred that *PgL0145810.1* may regulate the formation of seed hardness, which enhances the understanding of the function of the *SUT* genes in pomegranate.

Expression analysis indicated that the expression level of the *PgL0145810.1* gene varied in different tissues of transgenic lines, and most prominently in the stem part. Our result is similar to that, and the accumulation of sucrose is found more in mature, fully elongated internodes (at basal rather than the apical internodes) and the stems [57,58]. Additionally, the expression levels of *PgL0145810.1* in the stem of transgenic lines were higher than that of the wild type. Meanwhile, the lignin content of the stems in the transgenic *Arabidopsis* plants was significantly lower than in the wild type *Arabidopsis.* Cytological observation of the stem further verified that the length and the thickness of the xylem vessel in the transgenic lines were shorter and reduced as compared to the wild type plants. Previous studies proved that *PgL0145810.1* varied between soft-seeded cultivars and hard-seeded cultivars [13]. Thus, *PgL0145810.1* may play a role in the synthesis of lignin to regulate the structure of the xylem, thereby affecting seed hardness in pomegranate.

Promoter sequences that are located at the 5′ end are the DNA sequences responsible for the initiation of the transcription, with functions in the growth and development processes of plants. The promoter sequence in the navel orange sucrose synthase 1 (*CsSUS1*p) gene can drive foreign genes into the phloem which can be used to prevent and treat vascular bundle diseases [59]. The cis-regulatory elements within the specific gene reveal the gene expression patterns that are directly involved in environmental adaptation and further add to our understanding that they encode the proteins for plant growth and development, as well as fruit coloring in plants [60]. The MYB elements are typical cis-regulatory elements and have been proved to be related to the formation and evolution of important agronomic traits [61] and may be the main factors determining the pleiotropy of the *SUT* genes [20]. In this study, we identified different numbers of MYB cis-elements in the promoter region of *SUT* genes of pomegranate. *PgL0145810.1* has the highest number of MYB cis-elements in the promoter region among *SUT* genes in pomegranate. Global genome comparison analysis indicated that the gene sequence showed no difference between the comparisons: Tunisia_Dabenzi and Tunisia_Taishanhong [13]. Thus, we suppose that the MYB cis-elements in the promoter region may regulate *PgL0145810.1* expression to function in the development of seed hardness in pomegranate.

## 4. Materials and Methods

### 4.1. Plant Material

The pomegranate varieties, soft-seeded varieties (ZSL8, Yi3 and Tunisia) and hard-seeded varieties (Jiu Zi Hong, Yi2000-1 and Sanbai), used in this study were eleven-year-old trees grown in the nursery of the Zhengzhou Fruit Research Institute, Chinese Academy of Agricultural Sciences (CAAS). All of the pomegranate trees were robust with similar growth habits. The fruit samples from the trees were collected at 120 days after flowering (DAF). The seeds, along with the arils from the fruits, were extracted immediately and were placed in liquid nitrogen and finally frozen at −80 °C in the lab for further use.

### 4.2. Identification of SUT Genes from Pomegranate

We considered the whole genome sequence of pomegranate “Tunisia” [13] during our study. The *Arabidopsis SUT* gene sequences and protein sequences were downloaded from TAIR (https://www.arabidopsis.org/). The *SUT* gene family was downloaded from the TIGRFAMS database (J. Craig Venter Institute, La Jolla, CA, USA) [62]. We used the TIGR01301 [63] program to perform the sequence alignment and constructed a hidden Markov model (HMM) [64] for the structure of *SUT* proteins based on the alignment results. This HMM model was used to search the pomegranate protein database until the database was exhausted and, finally, pomegranate *SUT* protein sequences were compared with ClustalW. The seed sequences were used as the target sequences and the pomegranate protein sequences were analyzed in Blastp [65]. For the screening of the sequences, we considered E values less than 10–15 and got all the members of the *SUT* gene family.

### 4.3. Protein Sequence, Gene Structure and Promoter Analysis of SUT Genes

After the multiple sequence alignment through ClustalW, MEGA5.0 [66] was used to construct a phylogenetic tree with bootstrap set to 1000 replicates. Cluster analysis was performed on the protein sequences and promoter sequences of *SUT* genes. To obtain the intron numbers from each *SUT* gene, we extracted corresponding gff files. The 1.5kb upstream sequences of the genes were taken as the promoter sequences and we used the online tool PlantCARE (Flanders Institute for Biotechnology (VIB), Rijvisschestraat, Belgium) [67] to predict the cis-acting elements. To draw the gene structure and the cis-acting structure, we used the online tool GSDS 2.0 (http://gsds.cbi.pku.edu.cn/, GitHub, Inc., San Francisco, CA, USA) [68].

### 4.4. SUT Gene Protein Sequence Analysis and Isolation of PgL0145810.1

We used BioEdit software (https://bioedit.software.informer.com, BioEdit7.0.9, Informer Technologies, Inc., Carlsbad, CA, USA) [69] to analyze the pomegranate *SUT* genes and their upstream sequences and nucleic acid composition and calculate their GC content. The same software was used to analyze the amino acid composition and analysis of *SUT* protein molecular weight (*M*/*W*). The full-length sequence of *PgL0145810.1* was obtained from the whole genome sequence of pomegranate “Tunisia” from the seeds of both “Tunisia” and “Sanbai” through polymerase chain reaction (PCR). We used NCBI Primer Blast software (National Library of Medicine 8600 Rockville Pike, Bethesda MD, USA) [70] to design the primers (Appendix A).

### 4.5. Growth Conditions for Arabidopsis thaliana

The seeds of *A. thaliana* (ecotype Columbia) were washed with 95% ethanol (*v*/*v* in water) for 1 min for surface disinfection, followed with 50% sodium hypochlorite (*v*/*v* in water) for the next 3 min. Then, the seeds were subsequently rinsed five times with sterile deionized water under the clean bench. The disinfected seeds were placed in Petri dishes with 0.2X MS medium, pH 7.0 along with 3% sucrose and 1% plant agar. In the case of selecting the transgenic lines, the medium was supplemented with kanamycin (60 g/mL). For the uniform germination of the seeds, the Petri dishes were placed at 4 °C in the dark for 48 h and then transferred to the growth chamber. After seven days of germination, the plants were transferred to pots with substrate, top soil and perlite in the ratio 2:1:1 and were placed in the growth chamber with following conditions: 23–25 °C temperature, 70–80% relative humidity and 16:8 h light–dark photoperiod and 150 µmol m^−2^s^−1^ light intensity.

### 4.6. RNA Extraction and cDNA Synthesis

The total RNA from the pomegranate seeds was extracted using the cetyltrimethyl ammonium bromide (CTAB) method (Solarbio, Beijing, China). The cDNA was generated using one microgram of RNA by using a PrimeScript^TM^ RT Kit with gDNA Eraser (TaKaRa, Dalian, China) following the manufacturer’s protocol. A similar procedure was followed during the extraction of total RNA and cDNA from the transgenic lines of *Arabidopsis*.

### 4.7. Construction of the Vector and Genetic Transformation

The coding sequence (CDS) of the *PgL0145810.1* gene from pomegranate seeds was amplified from its cDNA by PCR. After amplification, the sequence was cloned into the pEASY-Blunt Simple vector. Then, it was further cloned into the PBI121-eGFP vector with XbaI/SacI restriction sites. The vector was transformed into *Agrobacterium tumefaciens* (GV3101 strain), then the *Agrobacterium* cell suspension was introduced into *A. thaliana* Col-0 inflorescences through the floral dip method [71,72]. The cell suspension contained 5% sucrose (*wt*/*vol*) and 0.05% Silwet L-77 (*vol*/*vol*) for an efficient absorption of the *Agrobacterium* by the female gametes [73,74]. After the floral dip, the plants were placed in the dark for 12 h and then transferred to light conditions in the growth chamber. Finally, the seeds were collected from the fully matured siliques.

### 4.8. Gene Expression Analysis through Quantitative Real-Time PCR (qRT-PCR)

Quantitative real-time PCR was conducted by using a Roche Light Cycler 480 Real Time PCR system and a Roche Light Cycler 480 SYBR Green I Master. The procedure during the qRT-PCR was as per the manufacturer’s protocol. The relative quantitative expression level was calculated by the 2^−∆∆^*^C^*^T^ method [75]. *Pg*Actin was used as an endogenous reference gene in sample tests to analyze the expression of genes in the different pomegranate varieties, and actin was used in the transgenic lines, performed in three replicates. The homozygous lines of transgenic three (T3) generations were used for the phenotypic expression analysis. The primers used are listed in Appendix A.

### 4.9. Sequencing of Genomic DNA

Total genomic DNA was isolated from young leaves of transgenic lines according to the procedures previously described [76]. The T-DNA insertion sites of the transgenic lines were examined by the genomic walking method [77,78]. The sequence of the right border region was used to design specific primers for thermal asymmetric interlaced PCR. The specific primers used during the study are listed in Appendix A.

### 4.10. Measurement of Lignin Content

The total lignin contents from the stems of the wild type *Arabidopsis* and the transgenic *Arabidopsis* plants were determined through the UV spectrophotometer method. The sample stems from the base of the plants for quantifying lignin were taken from 42-day-old seedlings. During the measurement of the lignin content, we used thioglycolic acid (TGA) assay procedures with slight modifications [79].

### 4.11. Cytological Analysis of Inflorescence Stem

For the cytological observation, the samples from the inflorescence stem of *Arabidopsis* (both wild type and transgenic) plants were taken at the 35-day-old seedling stage. The samples were placed in FAA solution (3.7% formaldehyde, 5% glacial acetic acid and 50% ethanol) under vacuum for 24 h for embedding with paraffin. The specimens were further sliced into ultrathin sections using an ultramicrotome and stained with 0.05% toluidine blue O and observed under an OLYMPUS BX51 light microscope (Hitachi, Ibaraki, Japan).

## 5. Conclusions

The whole genome sequence of pomegranate “Tunisia” was a reference genome from which we identified ten *SUT* genes. We performed genetic transformation, expression analysis and cytological observation of *PgL0145810.1*. The transgenic plants showed that the cells in the xylem were small and the lignin content was lower in the transgenic plants as compared to the wild type. This study provides a reference for understanding the *SUT* gene structure and their promoters as well as the expression of *PgL0145810.1* in transgenic plants.

## Figures and Tables

**Figure 1 ijms-21-06608-f001:**
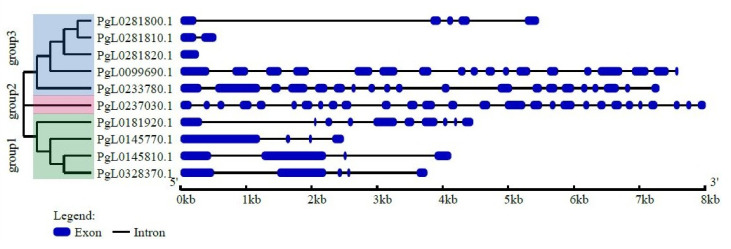
Phylogenetic tree of *SUT* genes of pomegranate and their exon/intron structure analysis. All pomegranate SUT protein sequences were determined with hidden Markov model (HMM) and were classified into three groups based on phylogenetic analysis. MEGA5.0 was used to construct a phylogenetic tree. The blue boxes represent exons and the black lines are introns.

**Figure 2 ijms-21-06608-f002:**
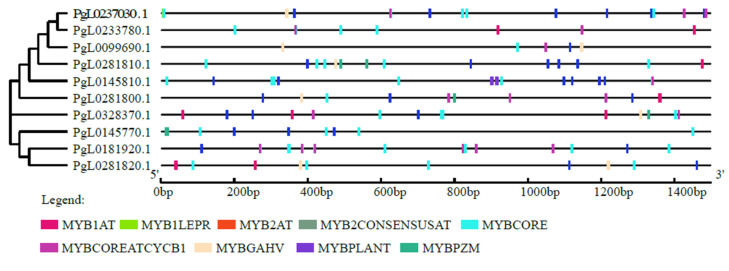
Promoters of *SUT* genes of pomegranate and their cis-acting element analysis. The online tool PlantCARE was used to predict the cis-acting elements and the Gene Structure Display Server (GSDS) 2.0 tool was used for constructing the cis-acting structure. The colored boxes represent different MYB elements.

**Figure 3 ijms-21-06608-f003:**
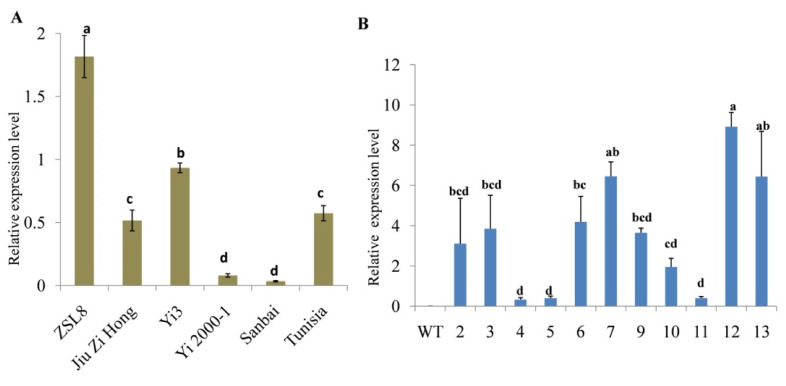
Relative expression level of the *PgL0145810.1*. Expression level of the gene in the seeds of different pomegranate varieties (**A**). Expression level of the gene in different transgenic *Arabidopsis* lines (**B**). Error bars indicate standard deviations from three replicates (*n* = 3). Values are means ± SD (*n* = 3). WT: Wild type and 2, 3, 4, 5, 6, 7, 9, 10, 11, 12 and 13: transgenic *Arabidopsis* lines. The letters indicate significant differences (*p* < 0.05).

**Figure 4 ijms-21-06608-f004:**
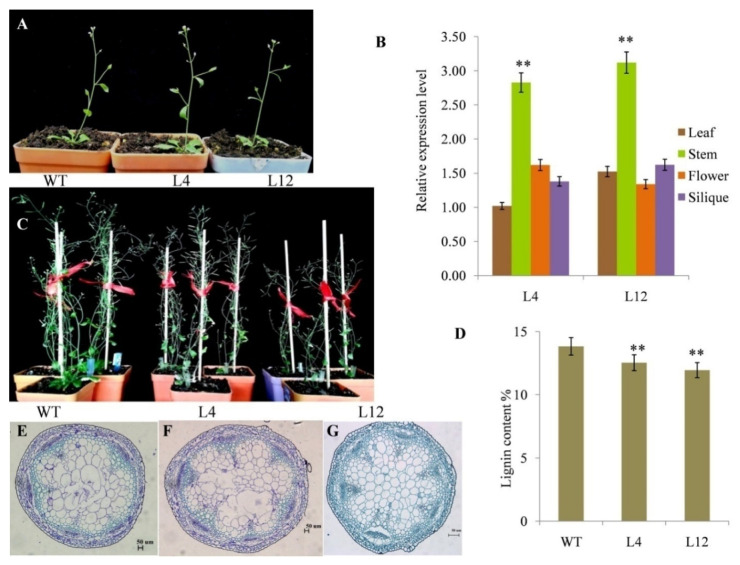
Growth parameters of wildtype (WT) *Arabidopsis* and transgenic lines (L4 and L12). Data shown are sample plants at the 35-day-old seedling stage for cytological observation (**A**), relative expression level of *PgL0145810.1* in different tissues of L4 and L12 (**B**), images of 8-week-old plants of WT, L4 and L12 (**C**), lignin content in the stems of WT, L4 and L12 (**D**), cytological observation of the flowering stems of WT, L4 and L12 (**E**–**G**). Error bars indicate standard deviations from three replicates (*n* =3). Values are means ± SD (*n*= 3). ** significant at *p* < 0.01 probability levels.

**Table 1 ijms-21-06608-t001:** Gene location, position, length and nucleotide composition of *SUT* genes in pomegranate.

Genename	Chr.	Start	End	Genelength (bp)	(A)%	(C)%	(G)%	(T)%	(C + G)%
*PgL0099690.1*	Chr1	2954359	2961954	7596	29.3	17.83	22.8	30.07	40.63
*PgL0145770.1*	Chr3	327678	330169	2492	28.18	22.84	24.97	24.01	47.81
*PgL0145810.1*	Chr3	341247	345377	4131	31.99	19.61	21.33	27.07	40.94
*PgL0181920.1*	Chr3	36608641	36613106	4466	34.13	21.59	19.89	24.39	41.48
*PgL0233780.1*	Chr4	39683636	39690941	7306	30.44	17.63	21.75	30.17	39.38
*PgL0237030.1*	Chr5	1480436	1488449	8014	30.38	21.3	17.97	30.35	39.27
*PgL0281820.1*	Chr6	13470881	13476350	5470	29.77	17.15	23.66	29.42	40.81
*PgL0281800.1*	Chr6	13476502	13477048	547	26.01	24.36	25.46	24.18	49.82
*PgL0281810.1*	Chr6	13470045	13470323	279	20.5	32.01	27.34	20.14	59.35
*PgL0328370.1*	Chr7	24075849	24079614	3766	24.33	21.46	22.55	31.66	44.01
Mean				4406.7	28.50	21.58	22.77	27.15	44.35

**Table 2 ijms-21-06608-t002:** Nucleotide composition of *SUT* promoters in pomegranate.

Gene_name	Chr.	(A)%	(C)%	(G)%	(T)%	(C + G)%
*PgL0099690.1*	Chr1	33.55	17.93	15.53	32.99	33.46
*PgL0145770.1*	Chr3	34.53	15.93	19.13	30.41	35.06
*PgL0145810.1*	Chr3	29.93	16.47	22.6	31	39.07
*PgL0181920.1*	Chr3	28.13	23.27	23.8	24.8	47.07
*PgL0233780.1*	Chr4	26.93	21.93	21.4	29.74	43.33
*PgL0237030.1*	Chr5	29.67	22.4	20.13	27.8	42.53
*PgL0281820.1*	Chr6	28.67	20.67	20.8	29.86	41.47
*PgL0281800.1*	Chr6	30.2	17.2	24.27	28.33	41.47
*PgL0281810.1*	Chr6	34	16	11.93	38.07	27.93
*PgL0328370.1*	Chr7	29.47	19.87	16.6	34.06	36.47
Mean		30.51	19.17	19.62	30.71	38.79

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
