# Peer review of "Identification of the SUT Gene Family in Pomegranate (Punica granatum L.) and Functional Analysis of PgL0145810.1"

_ijms, 2020, doi:10.3390/ijms21186608_

Round 1
Reviewer 1 Report
Dear Authors!
Your manuscript got better after making a few corrections that I recommended to you. Thank you for that. However, I must say that these adjustments were not major. Several important questions remain. I will repeat them in this review and hope that you will study them carefully and improve the quality of the manuscript in the new editorial circle.
- In the 2.1. part, you provide data of the GC composition analysis into the studied genes. The reason for this analysis is completely unclear to me. A statement about the GC composition effect on the stability of the molecular structure in a nucleic acid is speculative information. It requires evidence. The GC analysis is a clear departure from the main topic of the article. If you have a different opinion about this, then write your detailed comments for me.
- In the 2.1. and 2.2 part, you write about “GC content/gene length” and “histidine or methionine content/protein length” correlations. What is the biological meaning of these correlations? The scientific value of these results is extremely low. This unimportant information distracts from the main topic of the article. If you have a different opinion about this, then write your detailed comments for me.
- In the 2.6 and 2.7., you write about the PgL0145810.1 gene effects in the transgenic Arabidopsis lines. However, you do not indicate where the transgenes are embedded. The insertion site of transgenes must be studied to exclude the influence of the transgene position. For example, the transgene could damage the Arabidopsis genes, which are involved in the formation of xylem. I highly recommend to examine the transgene insertion sites of the resulting lines. I urge you to conduct a simple experiment to determine the place of transgene insertion into the genome in your Arabidopsis lines. This can be done using one of the chromosome walk methods, for example AL-PCR (Zheng SJ, Henken B., Sofiari E., Jacobsen E., Krens FA, KikC. Molecular characterization of transgenic shallots (Allium cepa L.) by adapter ligation PCR (AL-PCR) and sequencing of genomic DNA flanking T-DNA borders Transgenic Research, 2001. V. 10. P.237-245.). I think that information about the places of insertions will be more valuable than the results of the GC composition or “GC content/gene length” and “histidine or methionine content/protein length” correlation studying.
I also propose to correct the title of the manuscript as
«Identification of the SUT gene family in pomegranate (Punica granatum L.) and functional analysis of PgL0145810.1.»
I wish you every success in further improving your article. With best regards.
Author Response
R: Reviewer’s comments and suggestions
A: Author’s response
R: 1. In the 2.1. part, you provide data of the GC composition analysis into the studied genes. The reason for this analysis is completely unclear to me. A statement about the GC composition effect on the stability of the molecular structure in a nucleic acid is speculative information. It requires evidence. The GC analysis is a clear departure from the main topic of the article. If you have a different opinion about this, then write your detailed comments for me.
R: 2. In the 2.1. and 2.2 part, you write about “GC content/gene length” and “histidine or methionine content/protein length” correlations. What is the biological meaning of these correlations? The scientific value of these results is extremely low. This unimportant information distracts from the main topic of the article. If you have a different opinion about this, then write your detailed comments for me.
A: 1 and 2. Thank you for your comments. We agree to your view and deleted the related information about GC content and amino acid category in 2.1 and 2.2 part in order to highlight the theme of the manuscript. Additionally, we combined the 2.1 and 2.2 part in the revised manuscript. Please refer to the revised manuscript in the part of “result” sub-heading 2.1.
R: 3. In the 2.6 and 2.7., you write about the PgL0145810.1 gene effects in the transgenic Arabidopsis lines. However, you do not indicate where the transgenes are embedded. The insertion site of transgenes must be studied to exclude the influence of the transgene position. For example, the transgene could damage the Arabidopsis genes, which are involved in the formation of xylem. I highly recommend to examine the transgene insertion sites of the resulting lines. I urge you to conduct a simple experiment to determine the place of transgene insertion into the genome in your Arabidopsis lines. This can be done using one of the chromosome walk methods, for example AL-PCR (Zheng SJ, Henken B., Sofiari E., Jacobsen E., Krens FA, KikC. Molecular characterization of transgenic shallots (Allium cepa L.) by adapter ligation PCR (AL-PCR) and sequencing of genomic DNA flanking T-DNA borders Transgenic Research, 2001. V. 10. P.237-245.). I think that information about the places of insertions will be more valuable than the results of the GC composition or “GC content/gene length” and “histidine or methionine content/protein length” correlation studying.
A: 3. Thank you for your valuable suggestions. As we have already performed the genomic walking method at the first response but didn't add to the body part of the revised manuscript. Now, we have incorporated the result of the experiment for genomic walking method in the revised manuscript. Our result indicated that the transgene insertion sites of the resulting lines is at Chr5: 4,809,575 - 4,810,941. Because the transgene lines show the similar phenotypes, thus it is inferred that the transgene will not damage the Arabidopsis genes. We added the related information in the part of “result” under the sub-heading 2.6 in the revised manuscript.
R: 4. I also propose to correct the title of the manuscript as
Identification of the SUT gene family in pomegranate (Punica granatum L.) and functional analysis of PgL0145810.1.
A: 4. Thank you for your suggestions. We have changed the title of the manuscript as per your suggestions; “Identification of the SUT gene family in pomegranate (Punica granatum L.) and functional analysis of PgL0145810.1”.
Reviewer 2 Report
In the revised version of manuscript the Authors provide necessary information and improvements.
Author Response
R: Reviewer’s comments and suggestions
A: Author’s response
R: 1. In the revised version of manuscript the Authors provide necessary information and improvements.
A: 1. Thank you for your encourage and support to our response in the revised manuscript.
Round 2
Reviewer 1 Report
Dear authors! I see that a quality of your article have been incrеased. I will write my recommedation for publication. Thank you for your work!
With best regards.